# Routing Protocol for Intelligent Unmanned Cluster Network Based on Node Energy Consumption and Mobility Optimization

**DOI:** 10.3390/s25020500

**Published:** 2025-01-16

**Authors:** He Dong, Baoguo Yu, Wanqing Wu

**Affiliations:** State Key Laboratory of Satellite Navigation System and Equipment Technology, The 54th Research Institute, China Electronics Technology Group Corporation (CETC), Shijiazhuang 050081, China; yubg@sina.cn (B.Y.); wwqcetc@sina.cn (W.W.)

**Keywords:** high dynamic scenarios, intelligent unmanned clusters, routing protocol

## Abstract

Intelligent unmanned clusters have played a crucial role in military reconnaissance, disaster rescue, border patrol, and other domains. Nevertheless, due to factors such as multipath propagation, electromagnetic interference, and frequency band congestion in high dynamic scenarios, unmanned cluster networks experience frequent topology changes and severe spectrum limitations, which hinder the provision of connected, elastic and autonomous network support for data interaction among unmanned aerial vehicle (UAV) nodes. To address the conflict between the demand for reliable data transmission and the limited network resources, this paper proposes an AODV routing protocol based on node energy consumption and mobility optimization (AODV-EM) from the perspective of network routing protocols. This protocol introduces two routing metrics: node energy based on node degree balancing and relative node mobility, to comprehensively account for both the balance of network node load and the stability of network links. The experimental results demonstrate that the AODV-EM protocol exhibits better performance compared to traditional AODV protocol in unmanned cluster networks with dense node distribution and high mobility, which not only improves the efficiency of data transmission, but also ensures the reliability and stability of data transmission.

## 1. Introduction

With the advancement of low-cost, miniaturized sensor component technology and high-performance processor technology, miniaturized UAVs have experienced rapid development and widespread application in recent years [1,2]. Clustering, autonomy, intelligence, and systematization not only mark the forefront trends of UAV technology but also open up new fields and prospects for the practical application of UAVs [3]. Especially in domains such as forest fire prevention and monitoring [4], agricultural pest and disease surveillance [5], seismic disaster communication relay [6], battlefield intelligence reconnaissance [7], and target tracking in complex environments [8], intelligent unmanned clusters demonstrate their unique advantages.

The core of unmanned cluster networks lies in the interconnection and intercommunication of information among numerous UAV nodes through autonomous distributed network structure [9,10,11]. Compared to traditional information transmission networks, unmanned cluster networks exhibit superior flexibility, maneuverability and adaptability, supporting rapid deployment and movement of UAV nodes, eliminating the reliance on ground-based stations, and providing robust information exchange guarantees for the execution of unmanned cluster missions [12].

One of the typical application scenarios for contemporary unmanned clusters is in terrestrial urban environments, which has characteristics such as dense high-rise buildings, crisscrossing streets and complex underground infrastructure [13]. When unmanned clusters carry out tasks in complex urban environments with high dynamics and strong confrontation, there will be some problems such as severe limitations in spectrum resources [14] and intense competition for network resources [15]. The finiteness of network resources and the instability of network links can greatly affect the success rate of unmanned cluster task execution [16].

In view of the above problems, this paper focuses on how to ensure the link stability and data transmission reliability of unmanned cluster networks in highly dynamic and complex urban environments. This paper aims to propose an innovative routing protocol from the network level to help unmanned cluster networks adapt to the optimal routing path in highly dynamic environments, so as to reduce network congestion, minimize communication delay, and improve the overall efficiency and reliability of the network. The network routing protocol is fundamental to ensuring efficient data communication among UAV nodes in dynamic and challenging environments. Over the years, a variety of routing protocols have been developed to address the unique demands of unmanned cluster networks, such as high mobility, limited resources, and dynamic topologies. For instance, an intelligent routing protocol for multi-UAV networks was proposed, in which routing decisions are dynamically adjusted to ensure a minimum number of hops to the destination and better link quality by employing the Q-learning technique, resulting in improved end-to-end delivery and packet delivery ratio compared to the standard ad hoc routing protocol [17]. Similarly, Peng et al. introduced the FNTAR protocol, which makes routing decisions based on UAV future locations using trajectory, position, and motion data. This method reduces data loss in unstable links and dynamic topologies by forwarding messages to multiple optimal next-hop UAVs, outperforming the DTN geographic protocol in terms of latency and delivery ratio [18]. Moreover, a robust route discovery process based on the Penguins Search Optimization Algorithm (PeSOA) was proposed, designed to optimize routing paths by considering energy consumption and path connectivity, addressing challenges such as fast movement, packet loss, and poor communication in FANETs [19].

Although significant progress has been made in the design of routing protocols, challenges persist in adapting them to highly dynamic and complex urban environments. Urban areas, with their dense high-rise buildings and obstructions, further complicate the task of maintaining stable communication links. Therefore, there is increasing interest in developing routing protocols that can sense network conditions and dynamically adjust routing paths in real time. In order to ensure the real-time and accurate data transmission of unmanned cluster networks in complex urban environments, the AODV-EM protocol proposed in this paper comprehensively considers two routing metrics, network node energy consumption based on node degree balance and network node relative mobility, as the criteria for routing selection. According to the network link status and data transmission requirements, this protocol can plan appropriate transmission links for the data exchange among nodes, so as to reduce the amount of data forwarding among nodes and minimize the network routing cost. The significance of this study lies in its potential to fundamentally transform the application of unmanned clusters in complex environments, significantly improving data transmission efficiency and task success rate.

The rest of this paper is organized as follows. Section 2 describes the application scenarios and current issues. Section 3 proposes two routing metrics. In Section 4, the frame structure and workflow of the AODV-EM protocol are designed. The experimental setups and results that validate our approach are presented in Section 5. Finally, conclusions are drawn in Section 6.

## 2. Problem Description

### 2.1. Application Scenario Description

This paper mainly focuses on addressing the significant decline in collaborative positioning accuracy of intelligent unmanned clusters in high dynamic and strong interference environments. The actual application scenario is shown in Figure 1. In such environments, due to the refusal of satellite guidance, UAVs cannot use GPS to achieve autonomous positioning [20,21,22], so they need to be equipped with multi-source sensors such as UWB [23], IMU [24] and visual camera [25]. Due to the limited payload capacity of miniaturized UAVs, which results in generally inadequate sensor performance and susceptibility to obstacles and electromagnetic interference, individual UAVs face significant challenges in autonomous positioning, making collaborative positioning technology essential for accurate positioning and autonomous task execution in intelligent unmanned clusters [26,27].

Achieving accurate and reliable collaborative positioning of intelligent unmanned clusters depends on the interaction of position and status information among UAV nodes [28]. However, due to the high-speed movement of UAV nodes and interference from external environment factors, the frequent on-off of the unmanned cluster network link will cause the loss of transmission data [29], resulting in a significant decline in the collaborative positioning accuracy of the unmanned clusters [30]. Therefore, in order to ensure the real-time and accurate transmission of position and status information in the unmanned cluster network, it is necessary to design an intelligent unmanned cluster network routing protocol, which should be able to plan appropriate transmission links based on the network link status for positioning and navigation information transmission under conditions of high mobility and high load, reduce the number of information forwarding among nodes, minimize network routing overhead, and improve the overall information transmission quality of the unmanned cluster network [31,32].

### 2.2. Limitations of AODV Protocol

As a typical on-demand routing protocol, AODV does not need periodic maintenance of routing table [33,34], and has the characteristics of low latency, scalability, and non-centralization [35]. Although it has been widely used in unmanned cluster networks in recent years, there are still many problems with it, which can be summarized as follows:(1)As a passive routing protocol, AODV only initiates the routing process when nodes have communication requirements, and the process of establishing new routes requires a series of route discovery and maintenance operations [36].(2)In the process of AODV routing discovery, each node forwards the RREQ data frame to all its neighbor nodes within one hop [37]. However, when the number of nodes in the network is large or the density of nodes is substantial, this flooding data forwarding mode will lead to a significant increase in the number of data frames forwarded in a short time, which not only leads to the waste of network resources but also increases the risk of network congestion.(3)AODV protocol only takes “minimum hops” as the basis for route selection [38], and the data frames such as RREQ and RREP defined by AODV only store the information of source node, destination node and hop counts [39], without considering factors such as node mobility, link quality and node remaining energy. Therefore, AODV lacks the perception of the actual conditions of the network, and, in many cases, the selected shortest path may not be the optimal path for data transmission, thereby affecting the data transmission efficiency of nodes.(4)In AODV protocol, the key nodes in the network undertake more data transmission and forwarding tasks than the edge nodes due to their higher node degree, so key nodes experience faster energy consumption and a shorter lifetime. The failure of key nodes in the network may lead to widespread link disruptions, significantly reducing the network’s reliability and stability.

To solve the above problems, this paper proposes an intelligent unmanned cluster network routing protocol based on node energy consumption and mobility optimization to mitigate the drawbacks of traditional AODV protocol in applications and improve the performance of unmanned cluster networks.

## 3. Routing Metrics Design

### 3.1. Energy Consumption of Network Nodes

Assuming that the initial energy of all UAV nodes in the unmanned cluster network is $E_{start}$, the lifetime of network nodes is only related to their energy consumption without considering the damage caused by enemy attacks. The faster the energy consumption of a network node, the shorter its lifetime. Therefore, if the remaining energy of a node in the network is 0, it indicates that this node has exhausted its energy and will be separated from the network, unable to participate in subsequent data forwarding, and the network link connected to this node will also be broken. The remaining energy $E_{left}$ of network node is calculated as follows:(1)$$E_{left}=E_{start}-P_{idle}\times t-\sum_{N_{1}}E_{send}-\sum_{N_{2}}E_{recv}$$
where $P_{idle}$ represents the power consumption of the node in the idle state and t indicates the current survival time of the node. $E_{send}$ and $E_{recv}$ are the energy consumption for nodes when sending and receiving messages, respectively. $N_{1}$ and $N_{2}$ denote the number of messages sent and received by the nodes within the time period t.

When calculating the energy loss of unmanned cluster network nodes for sending and receiving data packets, the energy loss model of node communication adopted in this paper is shown in Figure 2.

The energy loss of the source node when sending a k-bit data packet to the destination node at a distance of d is as follows:(2)$$E_{send}(k,d)=\left\{\begin{array}{c} k\times E_{elec}+k\times \varepsilon_{fs}\times d^{2},d<d_{0}\\ k\times E_{elec}+k\times \varepsilon_{amp}\times d^{4},d\geq d_{0} \end{array}\right.$$
where $E_{elec}$ represents the energy consumed by the node to send unit bit data (unit is J), $\varepsilon_{fs}$ denotes the energy loss coefficient under the free space model, and $\varepsilon_{amp}$ is the energy loss coefficient under the multipath attenuation model, with the distance threshold $d_{0}=\sqrt{\varepsilon_{fs}/\varepsilon_{amp}}$.

The energy loss of the destination node when receiving a *k*-bit data packet is:(3)$$E_{recv}(k)=k\times E_{elec}$$
where $E_{elec}$ represents the energy consumed by nodes to receive unit bit data (unit is J).

Then, the energy loss rate $C_{i}$ of any node i in the network is as follows:(4)$$\begin{array}{ll} C_{i} & =1-\eta_{i}\\ & =\left(1-\frac{E_{i\_left}}{E_{i\_start}}\right)\times 100\%\\ & =\frac{E_{i\_start}-\left(E_{i\_start}-P_{idle}\times t_{i}-\sum_{N_{1}}E_{i\_send}-\sum_{N_{2}}E_{i\_recv}\right)}{E_{i\_start}}\times 100\%\\ & =\frac{P_{idle}\times t_{i}+\sum_{N_{1}}E_{i\_send}+\sum_{N_{2}}E_{i\_recv}}{E_{i\_start}}\times 100\% \end{array}$$
where $\eta_{i}$ denotes the energy residual rate of node *i*, $E_{i\_start}$ represents the initial energy of node *i*, $E_{i\_left}$ is the remaining energy of node *i* at this moment.

Thus, it can be seen that the more times a node participates in message forwarding in the network, the more energy it consumes, and the greater the energy loss rate becomes. However, when measuring the lifetime of network nodes, it is not accurate to only rely on the energy loss rate of nodes. If a node in the network exhibits a high energy loss rate but is situated at the network’s periphery, resulting in fewer subsequent message forwarding opportunities, its overall lifetime will become longer. Therefore, this paper introduces the concept of node degree in order to accurately judge the lifetime of network nodes. When the degree of a node in the network exceeds the average degree of the network nodes, it indicates that this node has more one-hop neighbors and will consequently undertake more message forwarding tasks, leading to an increasing trend in energy depletion. Conversely, if a node’s degree is below the average, the energy depletion trend will decrease.

In order to comprehensively consider the current energy loss and the subsequent energy loss trend of network nodes, this paper proposes an energy consumption evaluation index of network nodes based on node degree balance, which can be expressed as:(5)$$\begin{array}{ll} \bar{C_{i}} & =\ln(d_{i}+1)\times e^{C_{i}}\\ & =\ln(d_{i}+1)\times e^{\frac{P_{idle}\times t_{i}+\sum_{N_{1}}E_{i\_send}+\sum_{N_{2}}E_{i\_recv}}{E_{i\_start}}} \end{array}$$
where $d_{i}$ represents the node degree of node *i*, whose value is equal to the number of one-hop neighbors around node *i*, and $C_{i}$ denotes the energy loss rate of node *i*.

In order to balance the energy consumption of each node in the network as much as possible, it is necessary to prevent nodes with large energy consumption from participating in excessive data forwarding services as relay nodes. Therefore, the energy consumption evaluation index of network nodes based on node degree balance $\bar{C_{i}}$ proposed in this paper comprehensively considers the two factors of node degree and node energy loss, and emphasizes energy loss as the primary influencing factor through exponential amplification, while node degree is treated as the secondary influencing factor via logarithmic reduction. If the value of $\bar{C_{i}}$ is larger, the cost for node i to maintain the current data forwarding status is greater and the lifetime is shorter. Therefore, it is necessary to avoid this node as much as possible in subsequent data forwarding.

### 3.2. Relative Mobility of Network Nodes

In the unmanned cluster network, each UAV node is always moving at a high speed, and the relative position between the nodes is changing rapidly, which leads to the frequent occurrence of network link breakage. Therefore, the relative mobility of nodes at both ends of the link should be fully considered in the establishment of routing in the unmanned cluster network, and the motion state of the two nodes should be as close as possible to ensure the reliability and stability of information transmission to the greatest extent.

Each node in the unmanned cluster network is equipped with multi-source sensors, which allow it to obtain its own position and speed information in real time during the movement process. Therefore, the relative moving speed $v_{i,j}$ between any two nodes i and j in the network can be expressed as:(6)$$v_{i,j}=\sqrt{{\left(v_{i\_x}-v_{j\_x}\right)}^{2}+{\left(v_{i\_y}-v_{j\_y}\right)}^{2}+{\left(v_{i\_z}-v_{j\_z}\right)}^{2}}$$
where $v_{i\_x}$, $v_{i\_y}$ and $v_{i\_z}$ are the velocity components of node *i* in the x, y and z axes, respectively; $v_{j\_x}$, $v_{j\_y}$ and $v_{j\_z}$ are the velocity components of node *j* in the x, y and z axes, respectively. Each node is unified in the same reference coordinate system when calculating the position and speed information, and the speed component is positive when flying along the axis direction, otherwise it is negative.

To more effectively measure the relative motion difference between nodes, this paper introduces the concept of relative mobility, expressed as follows:(7)$$r_{i,j}=\frac{v_{i,j}}{\left|v_{i}\right|}=\frac{\sqrt{{\left(v_{i\_x}-v_{j\_x}\right)}^{2}+{\left(v_{i\_y}-v_{j\_y}\right)}^{2}+{\left(v_{i\_z}-v_{j\_z}\right)}^{2}}}{\sqrt{{\left(v_{i\_x}\right)}^{2}+{\left(v_{i\_y}\right)}^{2}+{\left(v_{i\_z}\right)}^{2}}}$$
where $r_{i,j}$ represents the relative mobility between nodes *i* and *j*, $v_{i,j}$ denotes the relative moving speed between nodes *i* and *j*, and $\left|v_{i}\right|$ indicates the absolute speed of node *i*.

In order to simplify the subsequent model calculation and network simulation, this paper focuses on the topological changes in the unmanned cluster network in a two-dimensional plane. Therefore, when calculating the relative mobility between nodes, the velocity component of nodes in the *z*-axis direction will not be considered, and the relative mobility $r_{i,j}$ can be expressed in the following form:(8)$$r_{i,j}=\frac{v_{i,j}}{\left|v_{i}\right|}=\frac{\sqrt{{\left(v_{i\_x}-v_{j\_x}\right)}^{2}+{\left(v_{i\_y}-v_{j\_y}\right)}^{2}}}{\sqrt{{\left(v_{i\_x}\right)}^{2}+{\left(v_{i\_y}\right)}^{2}}}$$

If the relative mobility $r_{i,j}$ between nodes *i* and *j* is greater, it indicates that the current motion direction and motion state of the two nodes are too different, and the network link between two nodes is more unstable, so it is necessary to avoid this link for data transmission as far as possible. Conversely, a smaller relative mobility signifies that the motion direction and state of the nodes are similar, leading to a more stable link, which should be prioritized for communication.

## 4. AODV-EM Routing Protocol Design

### 4.1. Frame Structure Design

The traditional AODV protocol only takes “minimum hops” as the unique routing criterion in the process of route selection, because the defined RREQ and RREP data frame formats only contain the Hop Count field as the criterion for route selection. Although this structure design can simplify the routing process and reduce computational complexity, it leads to several issues for AODV protocol in unmanned cluster networks, such as uneven energy consumption among network nodes and unstable routing paths. Therefore, the AODV protocol has certain limitations in the application of unmanned cluster networks.

In order to improve the above problems, the AODV-EM protocol proposed in this paper modifies the data frame formats of RREQ and RRQP in the original AODV protocol and introduces two routing metrics proposed in Section 3 on the basis of the original data frame structure. In this way, the energy consumption level and relative mobility of the next hop node can be considered comprehensively in the process of route selection, so as to achieve a more balanced route selection. Figure 3 and Figure 4 show the RREQ and RREP data frame formats of the AODV-EM protocol, respectively. The green part in the figure represents the new information based on the original AODV frame format.

It can be seen that the RREQ and RREP data frames of the AODV-EM protocol add node energy loss, *x*-axis speed of node, *y*-axis speed of node, and route cost on the basis of the original data frame, so the frame lengths increase by 12 bytes each. RREQ data frames are mainly responsible for route discovery, while RREP data frames are mainly used for route reply. By adding this information to the data frames, nodes can obtain the movement speed information of the previous hop node when receiving RREQ or RREP data frames. By evaluating both the energy consumption of a node and the relative mobility with its previous hop node, the routing cost for communication between two nodes can be calculated as follows:(9)$$\begin{array}{ll} RC_{i,j} & =\alpha \times \bar{C_{j}}+\beta \times r_{i,j}\\ & =\alpha \times \bar{C_{j}}+(1-\alpha )\times r_{i,j} \end{array}$$
where $RC_{i,j}$ represents the routing cost from node *i* to node *j*, $\bar{C_{j}}$ denotes the energy consumption value of node j based on node degree balance, $r_{i,j}$ indicates the relative mobility between node *i* and node *j*, and α and β are the weight coefficients for the metrics of node energy consumption based on node degree balance and relative mobility between nodes, respectively. The sum of α and β is always equal to 1, and their specific values can be dynamically adjusted based on the network conditions. If nodes in the network exhibit slower mobility, indicating that relative mobility has a smaller impact on routing cost, then α is set to a higher value and β to a lower value. Conversely, if nodes exhibit faster mobility, α is decreased and β is increased.

During the process of routing from the source node to the destination node, multiple intermediate nodes are passed through. By calculating the routing cost between each pair of nodes and summing these costs, the total routing cost for the link between the source node and the destination node can be determined as follows:(10)$$RC_{total}=\sum_{i=1}^{N-1}RC_{i,j}=\sum_{i=1}^{N-1}\alpha \times \bar{C_{j}}+(1-\alpha )\times r_{i,j},\;j=i+1$$
where $RC_{total}$ represents the total routing cost between the source node and the destination node, N is the number of all nodes on the link between the source node and the destination node, $RC_{i,j}$ denotes the routing cost of the link between node *i* and node *j*, and node *j* is always the next hop of node *i*.

In the subsequent simulation process, the moving speed $v_{node}$ of network nodes is set within the range of 5 m/s–50 m/s, so the weight coefficient α can be expressed as follows:(11)$$\alpha =0.2+(\frac{50-v_{node}}{45})\times 0.3$$

It can be observed that the value of the weight coefficient α decreases with the increase in the node movement speed $v_{node}$, which indicates that the faster the node movement speed, the higher the influence weight of the node relative mobility on the calculation of the total routing cost. When $v_{node}$ is set to 5 m/s, indicating that the nodes in the network move slowly, and the weight coefficient α=0.5, both routing metrics have an equal impact on the total routing cost. In contrast, when $v_{node}$ is set to 50 m/s, reflecting nodes in the network move rapidly, and the weight coefficient α=0.2, the relative mobility between nodes becomes the primary influencing factor.

### 4.2. Workflow Design

The AODV-EM protocol proposed in this paper, by modifying the workflow of the original AODV protocol in the process of route discovery and route recovery, abandons the “minimum hop count” as the only criterion for route selection, and calculates the route cost to the next hop node according to the energy consumption level and relative mobility of the neighboring nodes, so as to ensure the minimum cost value of the selected route. Next, the implementation of the AODV-EM protocol in the process of route discovery and route reply is introduced in detail. The route discovery and route reply processes are illustrated in Figure 5.

The source node broadcasts the RREQ data frames during the route discovery process. At this time, the node energy consumption and route cost value in these data frames are set to the initial value of 0. When an intermediate node receives the RREQ data frame, this node must verify whether this is the first reception and whether a reverse route already exists. If so, then perform the following steps. First, calculate the energy consumption based on node degree balance according to Equation (5) and update the energy information in the RREQ data frame. Next, calculate the relative mobility with respect to the previous hop according to Equation (8) and update the speed information in the RREQ data frame. Finally, compute the route cost value relative to previous hop node using Equation (9) and accumulate it with the route cost value stored in the RREQ data frame. If the route cost value of the current link is less than the cached value in the routing table, the routing information in the routing table will be updated, otherwise the RREQ data frame will be discarded.

When receiving the RREQ data frame, each intermediate node will determine whether it is the destination node. If not, it will continue the above process to update the relevant information in the RREQ data frame and add up the route cost value, while also continuing to broadcast and forward the RREQ data frame. The route discovery process continues until the destination node is found. At this time, the route cost value stored in the RREQ data frame represents the total route cost of the communication link between the source node and the destination node. The route discovery process of AODV-EM protocol is shown in Figure 6.

The destination node replies the RREP data frames to the source node after receiving the RREQ data frames during the route reply process. When an intermediate node receives a RREP data frame, it first checks whether there is a forward route in the routing table. If so, it calculates and accumulates the route cost value of the entire link based on the data frame information and compares it with the route cost value stored in the routing table. If the newly computed route cost value is smaller, the routing table will be updated accordingly.

When receiving the RREP data frame, each intermediate node will determine whether it is the source node. If not, it will continue the above process to update the relevant information in the RREP data frame and send the RREP data frame along the reverse route unicast in the routing table. Until the source node is found, the route reply process ends, and the source node starts data transmission to the destination node. The route reply process of AODV-EM protocol is shown in Figure 7.

## 5. Experiments

### 5.1. Experimental Setup

This paper utilizes NS-3 (Network Simulation Version 3) network simulation software to simulate data transmission among nodes in an unmanned cluster network. Compared with other network simulation tools, such as NS-2, OMNeT++, and Simulink, NS-3 as an open-source simulation platform based on C++ and has significant advantages. First, NS-3 comes with a comprehensive protocol stack that supports a wide range of common network protocols, including TCP, UDP, Wi-Fi, LTE, and 5G, making it highly adaptable to various network scenarios and simulation requirements. Additionally, users have the flexibility to access, modify, and extend the source code within NS-3, enabling the creation of customized network protocols and simulation modules, thus offering greater flexibility and support for researchers. More importantly, NS-3 is capable of large-scale network simulations, supporting the modeling of network environments with hundreds or even thousands of nodes. When simulating large-scale unmanned cluster networks, NS-3 can efficiently manage simulation resources, optimize simulation strategies, and ensure both the efficiency and accuracy of the simulation process [40,41].

In the virtual network simulated by NS-3, elements such as nodes, links, protocol stacks, and other elements in the network topology are abstracted as various C++ classes, and their connection operations are abstracted into associations among different C++ objects through function calls. For example, network nodes are represented by the Node class, and each simulated node requires the installation of NetDevice, Channel, Protocol Stack, Packets, Application, and other elements to achieve normal communication functions. This object-oriented design approach allows users the flexibility to construct and customize the desired network topology and behavior.

In addition, NS-3 software also provides users with powerful tool support. In this software, the FlowMonitor module can be utilized to monitor the transmission status of various data streams in the network and collect performance indicators such as transmission rate and packet loss rate. Furthermore, it also supports visualization tools such as NetAnim and PyViz, which allow for the visualization of network topology, node movements and communication status, thereby facilitating a more intuitive observation of network behavior during simulations.

In order to verify the performance improvement of the AODV-EM protocol proposed in this paper compared with the traditional AODV protocol in unmanned cluster network applications, NS-3 software is used to build a 1 km × 1 km simulation scenario, in which 50–140 UAV nodes are deployed, and each node moves irregularly within a speed range of 5–50 m/s. As shown in Figure 8, the red dots represent the UAV nodes, and the green lines represent the network links formed between the nodes, through which the specified data can be transmitted. The detailed simulation parameters of NS3 software configured in this paper are shown in Table 1.

### 5.2. Simulation Results Analysis

In order to intuitively demonstrate the performance improvement of the proposed AODV-EM protocol, this paper conducts comparative simulation experiments against the traditional AODV, OLSR [42], and DSDV [43] protocols, evaluating five key metrics: packet delivery rate, network throughput, average end-to-end delay, average jitter, and standard deviation of node residual energy. To ensure the reliability of the simulation results, this paper conducted five independent experiments for each indicator, followed by a comprehensive analysis of the results. The simulation outcomes for each indicator are presented below.

(1) Comparison of packet delivery rate: Figure 9 presents the simulation results of four routing protocols in terms of packet delivery rate. Figure 9a illustrates that, as the number of nodes increases, the packet delivery rate for all protocols generally decreases. This decline is due to the growing complexity of the network topology, which leads to higher routing costs, including increased route discovery and maintenance overhead. As a result, network congestion rises, negatively impacting the packet delivery rate. Figure 9b shows the effect of node movement speed on the packet delivery rate. As node speed increases, the AODV-EM protocol outperforms the other three protocols, with the performance gap widening as node speed increases. This indicates that faster node movement leads to more frequent link breakages, destabilizing data transmission and reducing packet delivery rate. However, the AODV-EM protocol accounts for relative node mobility when establishing routes, resulting in more stable network links that are less susceptible to the effects of node movement. As a result, AODV-EM protocol maintains a higher packet delivery rate despite increased node speed.

(2) Comparison of network throughput: Figure 10 presents the simulation results of four routing protocols in terms of network throughput. From Figure 10a,b, it is clear that the AODV-EM protocol and the AODV protocol outperform the OLSR protocol and the DSDV protocol in terms of network throughput. This advantage is primarily due to the on-demand routing approach used by the AODV protocol and the AODV-EM protocol, where route discovery and updates occur only when data transmission is required. This significantly reduces unnecessary routing control overhead, thereby improving overall network throughput. As the number of nodes or node movement speed increases, the AODV-EM protocol demonstrates greater stability in network throughput compared to the AODV protocol, with a slower decline in performance. This is because the AODV-EM protocol takes into account the relative mobility between nodes when selecting routes, allowing it to better adapt to dynamic network conditions and minimize the impact of link breakages, making it more resilient to changes in the network topology.

(3) Comparison of average end-to-end delay: Figure 11 presents the simulation results comparing four protocols based on the average end-to-end delay. As shown in Figure 11a,b, the OLSR protocol, which continuously and periodically maintains network topology information, can quickly calculate the shortest path and update routes, achieving the best performance in terms of end-to-end delay. In contrast, the DSDV protocol relies on distance-vector-based updates, which have slower convergence and are more affected by control overhead and route selection delays, leading to the poorest performance in terms of average end-to-end delay. As a reactive routing protocol, the AODV protocol only initiates route maintenance when data transmission is required, resulting in a slightly higher average end-to-end delay than the OLSR protocol. The AODV-EM protocol proposed in this paper, however, takes into account the energy consumption and relative mobility of the next-hop node during the path-finding process, resulting in more stable network links and reduced delay from route maintenance. Its performance advantage is particularly evident in scenarios with frequent network topology changes.

(4) Comparison of average jitter: Figure 12 presents the simulation results of four protocols with respect to the average jitter metric. Average jitter reflects the fluctuation in delay during data transmission in the unmanned cluster network, while both average end-to-end delay and average jitter are indicators of the network link’s reliability and stability [44,45]. As a result, the simulation comparison curves for these two metrics exhibit similarities. The simulation results for average jitter further demonstrate that the AODV-EM protocol proposed in this paper offers superior stability in network links and more reliable data transmission, particularly under complex network topology conditions.

(5) Comparison of standard deviation of node residual energy: Figure 13 presents the simulation comparison results of four protocols in the standard deviation of node residual energy. As shown, the standard deviation of node residual energy of each protocol gradually increases with the advance of simulation time. The standard deviation reflects the degree of dispersion of the data distribution, indicating that the distribution of node energy in the network gradually becomes uneven. Notably, the OLSR and DSDV protocols perform worse due to their nature as proactive routing protocols, which require periodic control message exchanges even when no data transmission occurs [46,47]. This leads to more frequent and uneven energy consumption across the nodes. In contrast, the AODV-EM protocol exhibits a significantly slower increase in the residual energy standard deviation, with the gap between it and the AODV protocol widening over time. This is because the AODV protocol relies solely on "minimum hop count" for route selection, causing key nodes to bear a disproportionate amount of data forwarding tasks and deplete their energy more quickly. As these critical nodes run out of energy and go offline, network links can break, disrupting connectivity and data transmission. On the other hand, the AODV-EM protocol accounts for the energy levels of the next-hop nodes when selecting routes, prioritizing nodes with higher energy reserves. This strategy results in more balanced energy consumption, prolongs the node lifecycle, and enhances network link stability.

## 6. Conclusions

This paper proposes a network routing protocol, AODV-EM, which optimizes node energy consumption and mobility to address the need for real-time, accurate transmission of position and status information in unmanned cluster cooperative positioning networks, particularly under conditions of high mobility and heavy load. The NS-3 simulation platform is used to compare AODV-EM protocol with the traditional AODV, OLSR, and DSDV protocols in terms of packet delivery rate, network throughput, average end-to-end delay, average jitter, and standard deviation of node residual energy. The experimental results show that the AODV-EM protocol has better performance for unmanned cluster networks under conditions of dense node distribution and high node mobility, which not only improves the efficiency of data transmission, but also ensures the reliability and stability of data transmission. However, when several key nodes in the unmanned cluster network fail, the network link will be broken, and the network performance cannot be fundamentally improved by optimizing the routing protocol alone. Therefore, future research will focus on the design and implementation of network topology reconstruction algorithms. These algorithms will be combined with routing protocols after node failure to dynamically adjust network topology, restore connections and optimize data transmission paths, so as to effectively mitigate the negative impact of node failure on network performance.

## 7. Patents

The results of this will be submitted as patent within this year.

## Figures and Tables

**Figure 1 sensors-25-00500-f001:**
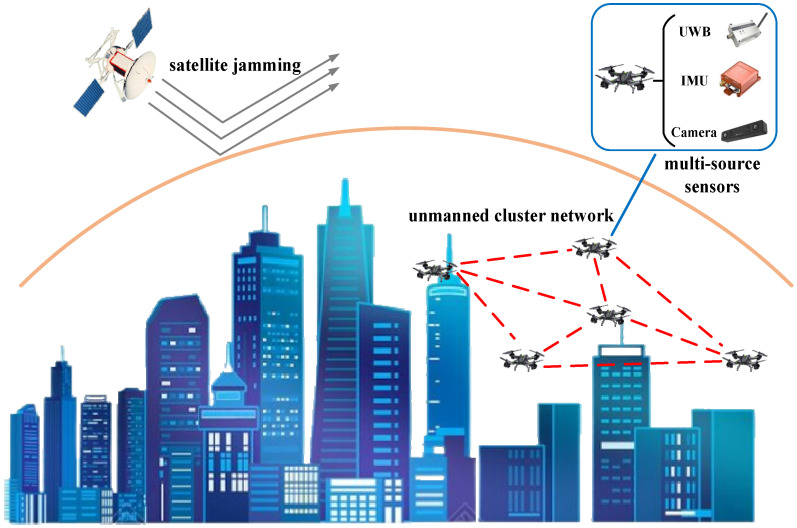
Unmanned cluster collaborative positioning network.

**Figure 2 sensors-25-00500-f002:**
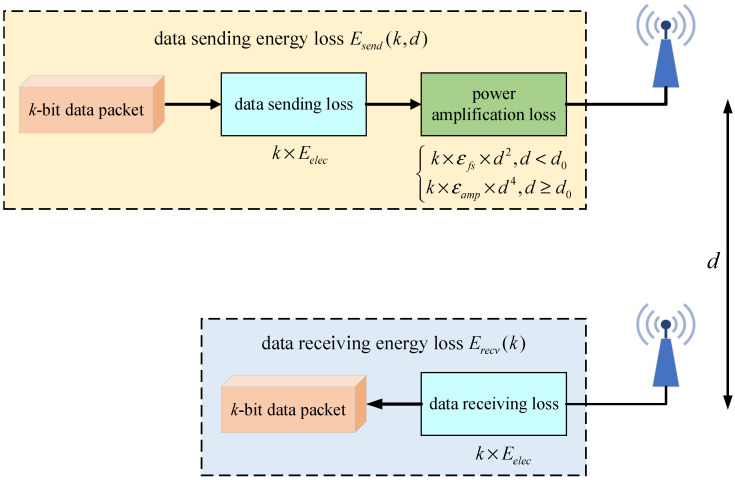
Network node communication energy loss model.

**Figure 3 sensors-25-00500-f003:**
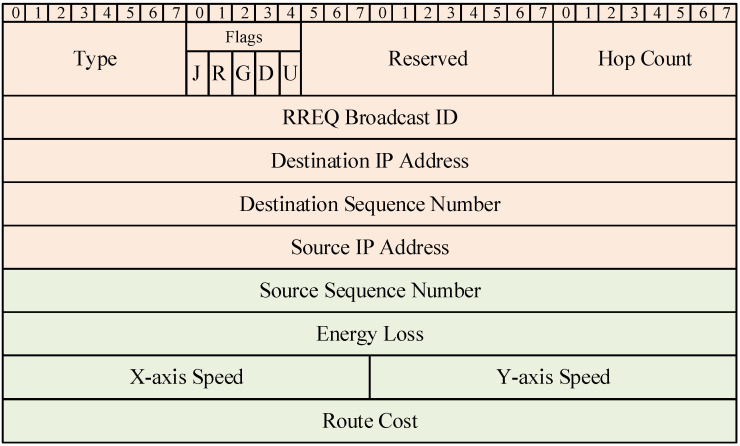
RREQ data frame format in AODV-EM protocol.

**Figure 4 sensors-25-00500-f004:**
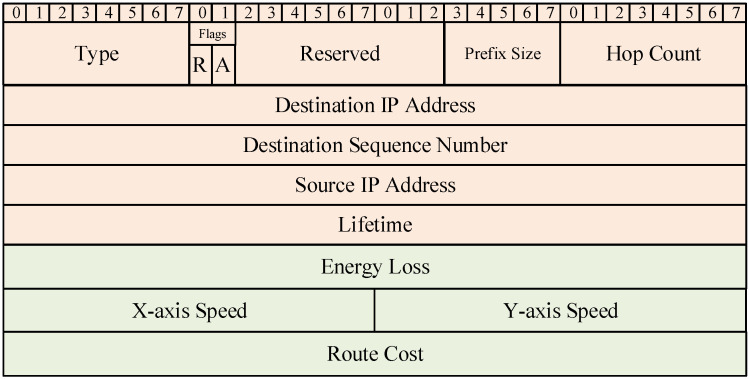
RREP data frame format in AODV-EM protocol.

**Figure 5 sensors-25-00500-f005:**
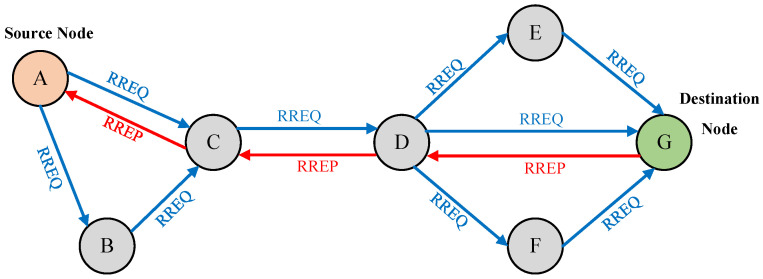
Route discovery and route reply processes.

**Figure 6 sensors-25-00500-f006:**
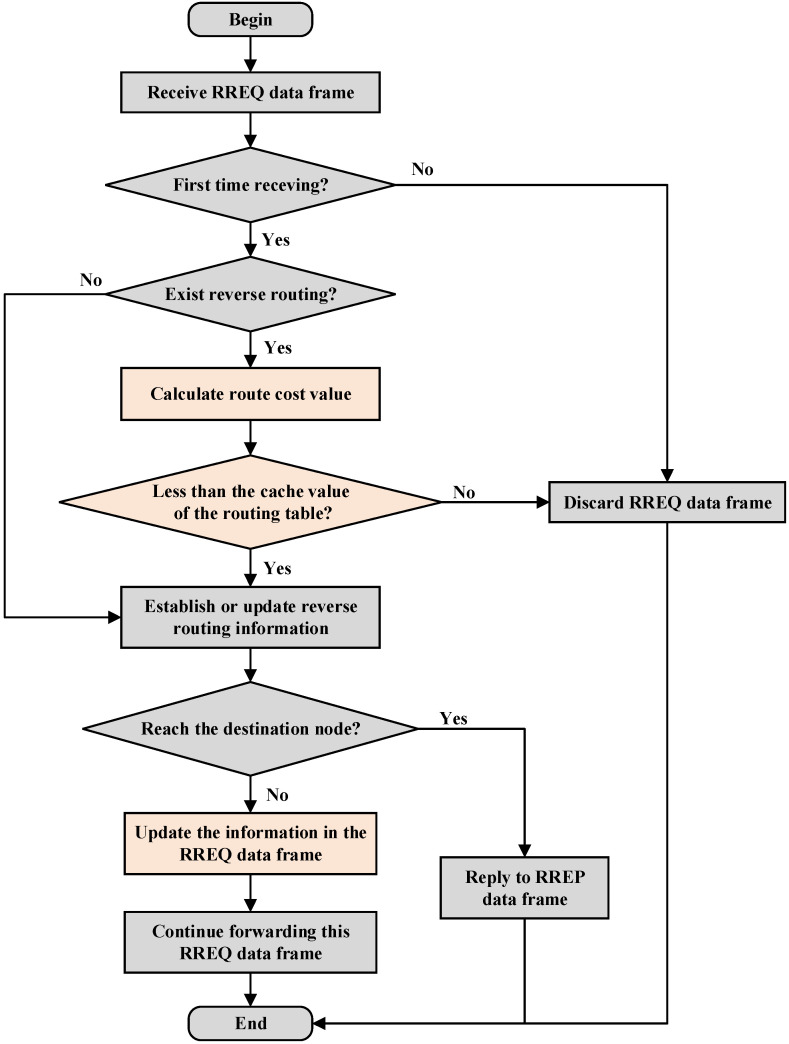
Route discovery process of AODV-EM protocol.

**Figure 7 sensors-25-00500-f007:**
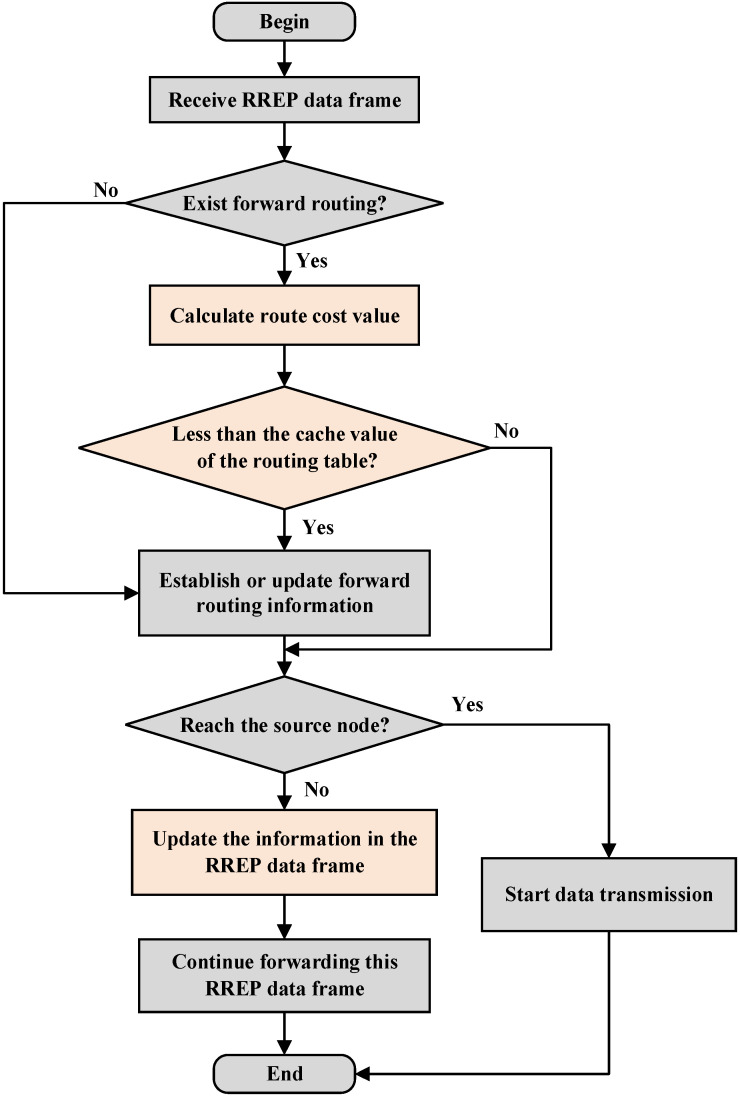
Route reply process of AODV-EM protocol.

**Figure 8 sensors-25-00500-f008:**
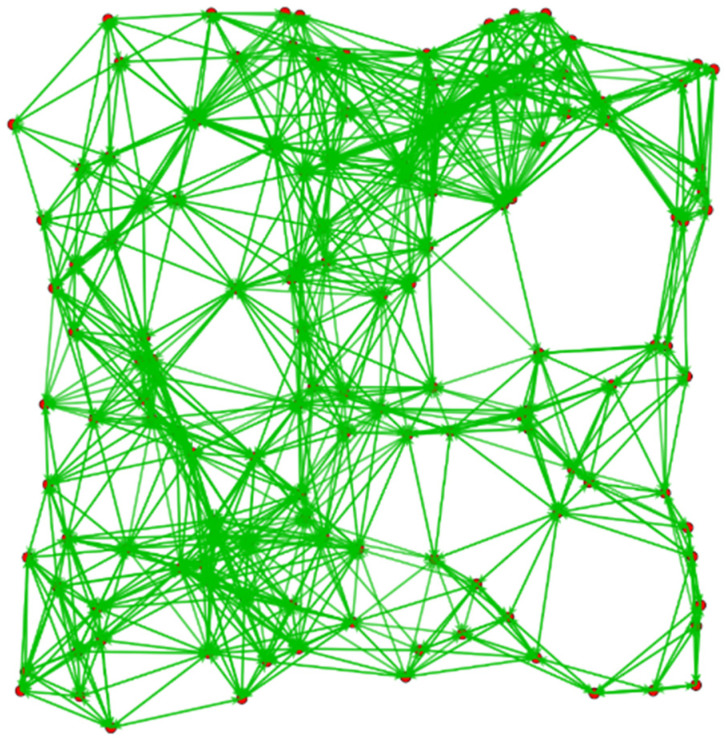
Simulation scenario built by NS-3 software.

**Figure 9 sensors-25-00500-f009:**
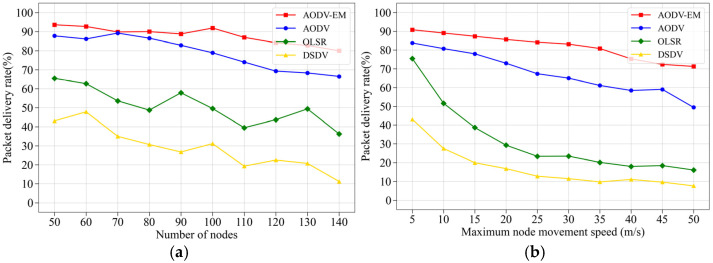
Comparison of packet delivery rate. (**a**) Variation in the number of nodes. (**b**) Variation in the maximum node speed.

**Figure 10 sensors-25-00500-f010:**
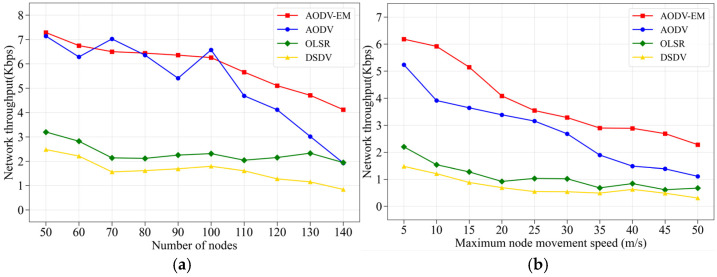
Comparison of network throughput. (**a**) Variation in the number of nodes. (**b**) Variation in the maximum node speed.

**Figure 11 sensors-25-00500-f011:**
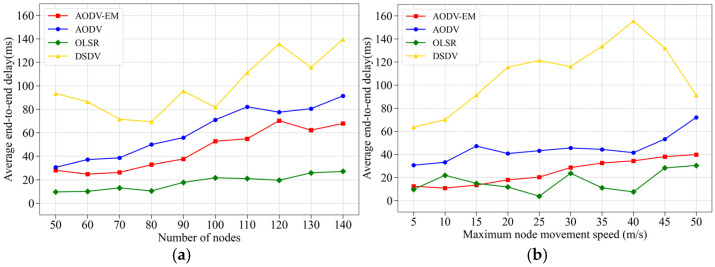
Comparison of average end-to-end delay. (**a**) Variation in the number of nodes. (**b**) Variation in the maximum node speed.

**Figure 12 sensors-25-00500-f012:**
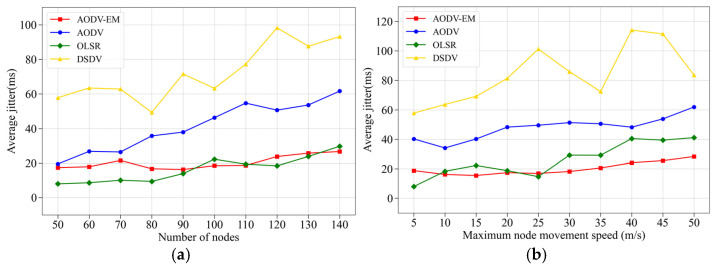
Comparison of average jitter. (**a**) Variation in the number of nodes. (**b**) Variation in the maximum node speed.

**Figure 13 sensors-25-00500-f013:**
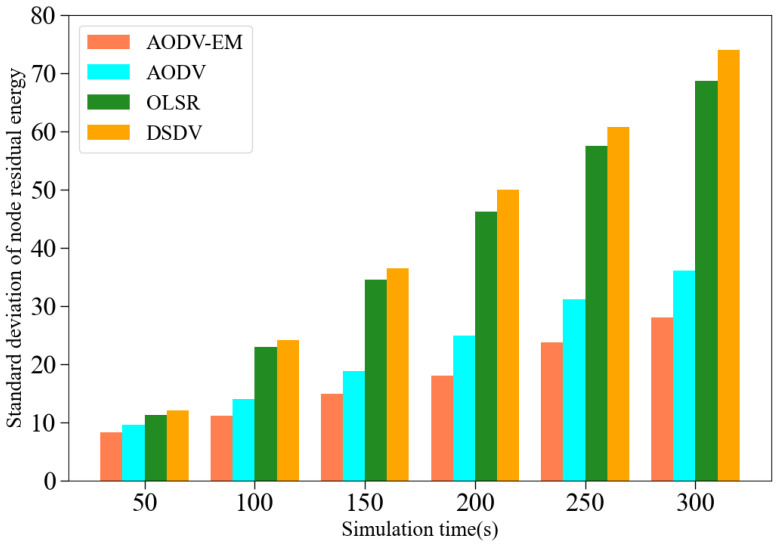
Comparison of standard deviation of node residual energy.

**Table 1 sensors-25-00500-t001:** Simulation parameters configuration.

Parameters Name	Parameters Value
Version	NS-3.28
Simulation Time	300 s
Simulation Scenario Scope	1000 m × 1000 m
Number of Nodes	50–140
Maximum Transmission Distance	200 m
Node Mobility Model	Random Waypoint Mobility (RWP)
Node Movement Speed	5–50 m/s
Packet Length	512 bits
Business Type	Constant Bit Rate (CBR)
Transport Layer Protocol	UDP
Mac Layer Protocol	IEEE 802.11b
Delay Model	Constant Propagation Delay Model
Node Initial Energy	200 J
Energy Consumption for Sending Data	0.05 J
Energy Consumption for Receiving Data	0.03 J

## Data Availability

The data presented in this study are not publicly available.

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
