# Peer review of "Routing Protocol for Intelligent Unmanned Cluster Network Based on Node Energy Consumption and Mobility Optimization"

_sensors, 2025, doi:10.3390/s25020500_

Round 1
Reviewer 1 Report
Comments and Suggestions for Authors
1. The article lacks a more intuitive schematic comparison between the AODV protocol and the AODV-EM protocol. It is recommended to add a schematic diagram to facilitate the distinction between the two and to demonstrate the superiority of the AODV-EM protocol.
2. The article only compares the AODV-EM protocol with the traditional AODV protocol, lacking comparisons with other improved AODV protocols or routing protocols suitable for highly dynamic networks. It is recommended that the authors add comparative experiments with more benchmark protocols, such as OLSR and DSR, to more comprehensively evaluate the performance of the AODV-EM protocol.
3. Figures 3 and 4 show the data frame format of the AODV-EM protocol. However, the modifications compared to AODV are not clearly indicated in the figures. It is suggested to annotate the modified parts more clearly or to retain the original AODV data frame format for a side-by-side comparison.
4. The article mentions that the experimental results demonstrate the superiority of the AODV-EM protocol, but it does not provide statistical analysis of the experimental data, such as confidence intervals or significance tests. It is recommended that the authors supplement statistical analysis to enhance the credibility of the experimental results.
5. The article does not address the security considerations of the AODV-EM protocol, such as how to deal with malicious nodes or attacks in the network. It is recommended that the authors discuss potential security issues of the protocol and propose corresponding improvement measures.
Comments on the Quality of English Language
English can be improved.
Reviewer 2 Report
Comments and Suggestions for Authors
This manuscript, "Routing Protocol for Intelligent Unmanned Cluster Network Based on Node Energy Consumption and Mobility Optimization," presents the role of intelligent unmanned clusters in various fields such as military reconnaissance, disaster rescue, and border patrol. The paper proposes an AODV routing protocol based on node energy consumption and mobility optimization (AODV-EM). The experimental results show that the AODV-EM protocol performs better than the traditional AODV protocol in dense and high-mobility unmanned cluster networks.
-
Authors must fully describe the introduction section, including the background, history, problem statement, objectives, scope, significance, and methodology overview.
-
Authors must fully describe the literature review on the subject and topics, but there is no literature section in the manuscript. Assume the readers are beginners.
-
Describe the NS-3 (Network Simulation Version 3) network simulation software to the reader. Why did the authors pick this simulation program? Are there any other simulation programs or comparisons? Did any different effects occur during the simulation?
-
Figure 11, (a) shows the red line going lower, but the blue line going higher at 100 nodes. (b) shows the red line going lower, but the blue line going higher at 20 nodes. These appear to be significant factors in the results of (a) and (b). Authors must describe the meaning of these graphs.
-
Figure 12, (a) shows the red line going lower, but the blue line going higher at 100 nodes. (b) shows the red line going lower, but the blue line going higher at 15 nodes. These appear to be significant factors in the results of (a) and (b). Authors must describe the meaning of these graphs.
-
This manuscript presents the experimental simulation well, but the conclusion lacks confidence in the experimental simulation results. I believe the authors can provide more details in the revised version.
Thank you. Well done.
Round 2
Reviewer 1 Report
Comments and Suggestions for Authors
There is no other problems.